# A Study on South Korean College Students’ Perceptions of Gratitude

**DOI:** 10.3390/bs13040281

**Published:** 2023-03-23

**Authors:** Namki Lee, Yucheon Kim

**Affiliations:** Department of Counseling and Coaching, Dongguk University, 30, Pildong–ro 1 gil, Jung–gu, Seoul 04620, Republic of Korea; nearman7@dgu.ac.kr

**Keywords:** college students, gratitude, Q methodology, subjectivity

## Abstract

Humans feel happy when they experience positive emotions; gratitude is a significant inducer of positive emotions. This study investigates perceptions of gratitude among South Korean college students using the Q methodology, which enables the examination of individuals’ subjectivity. We collected 227 statements from a Q population through literature reviews, paper reviews, interviews, and questionnaire surveys; from them statements, we selected 40 Q samples. The P samples included 46 college students at Dongguk University in Seoul, South Korea, and we performed data analysis with Principal Component Factor Analysis using the Quanl program. Using the results of this study, we classified gratitude into five types: Type 1 active gratitude through expression; Type 2 passive gratitude depending on conditions; Type 3 gratitude through relationships; Type 4 gratitude through internal satisfaction, and; Type 5 gratitude through materials. The results point to differences in experiences of gratitude that depend on conditions and environments, and by type. Researchers and administrators can use the results of this study to understand South Korean college students’ perspectives and perceptions when planning and implementing gratitude programs that prioritize their happiness.

## 1. Introduction

Positive psychology [1] focuses on human well-being, and studies how individuals can lead happy lives. It emphasizes that, rather than eliminating individuals’ negative emotions or personality weaknesses, generating positive emotions and highlighting strengths is more important for happiness [2]. We become happy when we have positive feelings about the past, present, and future [3]. Wilson [4] defined happiness as Subjective Well-Being (SWB); SWB occurs when one experiences more positive affect than negative affect [5]. In addition, researchers have found a high correlation between gratitude, a positive emotion, and life satisfaction [6]; it is an important element in measuring SWB [7].

Gratitude is a personality strength closely related to happiness, and people who experience gratitude show higher levels of positive emotions, life satisfaction, and optimism. In particular, gratitude has a larger effect in enhancing positive emotions than in reducing negative ones [8]. When researchers examined gratitude’s etymology, they interpreted it as a state of being grateful to others or a mind to repay a favor [9]. Expression of gratitude is the most basic positive activity in human society [10], and major religions, such as Buddhism, Islam, Judaism, and Christianity, have emphasized it as a core virtue for individuals [11].

Meanwhile, during their college days, students often complain of psychological difficulties stemming from their newfound independence from parents, interpersonal relationships, and academic and career choices [12]. In addition, college days correspond to early adulthood, and are one of the most lonely and challenging times in life. Many students receive little attention and help from surrounding people during college days, and use this period to begin solving important life tasks [13]. Therefore, we became interested in how college students perceive gratitude, which can promote positive emotions. To that end, this study focused on how college students perceive individuals’ SWB and gratitude, which plays a culturally significant role.

We reviewed studies on the effect of gratitude in helping students to overcome difficulties in college. For instance, Lee and Shim [14] found that gratitude has positive effects on students’ overall college life, such as improving academic achievement and attachment to the college. In addition, when college students write a gratitude diary, their ego resilience increases [15,16,17] thus, the more gratitude a person has, the more they feel happy and optimistic [10,18,19]. Regarding gratitude scales, Watkins et al. [20] developed the Gratitude, Resentment and Appreciation Test (GRAT) to separately measure gratitude influenced by subjective elements primarily in three factors: awareness of affluence; gratitude in daily life; and gratitude to others. Furthermore, McCullough et al. [8] selected six items to measure gratitude quantitatively. In addition, many researchers have conducted studies to understand the concept quantitatively by scaling gratitude [21,22,23,24,25].

Although researchers explored the effects and scales of gratitude as described above, studies examining the subjective perception of gratitude are insufficient. Furthermore, since individuals’ subjective perceptions affect how they interpret the surrounding environment [26], the perspective of gratitude perceived by individuals cannot be different. Therefore, this study explores the subjectivity of gratitude to fill the research gap.

Since gratitude is a product of the transformation of self-perspective to examine one’s self [27], the degree to which norms and emotions affect gratitude varies from person to person [28]. Q methodology enables the examination of individuals’ subjectivity for objects such as gratitude that one cannot define concretely, and which individuals perceive differently. Thus, this methodology is suitable for this study because it provides a foundation for investigating individuals’ opinions, views, attitudes, and beliefs about specific objects or contexts [17]. Q methodology is a form of research that focuses on the subjective inner experiences of individuals, measuring their preferences, emotions, ideals, tastes, and other related factors. It is a study of human beings that seeks to understand their unique perspectives and experiences. The first researcher to present the Q methodology was William Stephenson, who outlined this methodology in a paper on nature in 1935 [29]. It is a quantitative and qualitative mixed research methodology [30] that aims to understand individuals’ subjective concepts about special phenomena [31]. In this methodology, the study begins from the actor’s point of view instead of the researcher’s assumption; the statements used are self-referential opinion items [29]. This approach is not intended to verify a specific hypothesis; rather, it serves as an abductory methodology for generating and exploring potential hypotheses [32]. Q methodology uses factor and correlation analyses to statistically classify and analyze statements expressing humans’ subjective thoughts or feelings [33].

Therefore, this study uses the Q methodology to explore how college students perceive gratitude. The study analyzes the types of gratitude college students perceive to help them have a grateful mind, and thus live a happier life. The study will answer the following questions:

Research question 1: What are the types of gratitude perceived by college students?;

Research question 2: What are the differences in the characteristics of these types?

## 2. Study Method

Table 1 summarizes this study’s process.

### 2.1. Organization of Q Population

A Q population is a concourse of individuals’ subjective perceptions collected for a Q study. There are several methods, such as studies through related literature surveys, interviews, and questionnaire surveys, through which to create a Q population. In this study, we formed a Q population through a related literature survey, interviews, and questionnaire surveys. Regarding related literature, we selected five books with the most subscriber reviews among books on gratitude. We also extracted the study’s definitions and expressions of gratitude from the books (*n* = 58). In addition, we found papers that conducted interviews on gratitude, and extracted statements about gratitude from the papers (*n* = 13). For the interviews, we conducted semi-structured interviews with three college students on the theme of gratitude to extract statements (*n* = 32). Furthermore, we used a questionnaire to survey gratitude perceptions among 12 students (four college and eight graduate students, all majoring in psychology), using semi-structured questions to extract statements (*n* = 124). As a result, we organized 227 statements of a Q population through individual processes.

### 2.2. Selection of Q Samples

Q samples are statements extracted from the Q population. We listed the Q samples with the Q population’s 227 statements and classified them into the three gratitude factors of GRAT: awareness of affluence; gratitude in daily life, and; gratitude to others. We repeatedly analyzed the statements to remove duplicate items; we selected secondary samples through guidance by experts who lecture on Q methodology at graduate schools, and reviews by three doctoral students who had taken a course in Q methodology. In addition, we modified sentences that were ambiguous in meaning or incomprehensible through joint discussions. Through the preliminary test of the Q samples described above, we selected 40 final Q statements, as shown in Table 2. Finally, we classified the Q statements into awareness of affluence (*n* = 9), gratitude in daily life (*n* = 13), and gratitude to others (*n* = 18).

### 2.3. Composition of P Samples

P samples classify the extracted Q statements based on their subjective perceptions. The P samples in this study comprised 46 college students from Dongguk University in Seoul, South Korea. By grade, the P samples consisted of 13 first-graders, nine second-graders, ten third-graders, and 14 fourth-graders; there were 20 male and 26 female students. Before conducting the survey, we sufficiently explained the purpose and procedure of the study to the P samples, and obtained their prior consent to participate. In return for their study responses, we gave the participants coffee coupons.

### 2.4. Reliability Test

The test–retest reliability process is a method of measuring the reliability of the Q samples [33], with consistency verified through a pre-test before completing the Q samples [34]. Researchers usually perform the test with two to six persons, performing Q sort twice at a time interval to obtain the average correlation coefficient value [31]. Researchers generally consider reliability to be sufficiently high when r = 0.7 or higher [31]. In this study, to secure reliability through a reliability test, we repeated the Q sort at a one-week interval with three P samples, obtaining an average correlation coefficient value of 0.682 as shown in Table 3. Therefore, the test secured a relatively high reliability value that was close to 0.7, validating the final 40 Q samples.

### 2.5. Q Sorting and Data Analysis

After the reliability test, we performed Q-sorting, which involves P samples sorting Q samples. Before entering Q-sorting, we requested three doctoral students majoring in Q methodology to perform preliminary Q-sorting. We then completed a final check on the Q statements through the pre-Q-sorting work, and prepared the Q-sorting guidance work. Through this process, we prepared explanations of the study’s purpose, and additionally adjusted Q statements that were not smooth.

After a preliminary preparation, we presented 40 statement cards selected as Q samples to the P samples, who performed Q-sorting from 17 to 29 October 2022. The participants sorted the Q statement cards as positive, neutral, or negative. We then performed a forced distribution, so the 40 cards made a normal distribution. Based on the statement numbers recorded in the Q sample distribution chart, we ordered the statements from 1 point for the most disagreeable items to 11 points for the most agreeable items. We then coded the scored list, entered it into the Quanl program, and analyzed and interpreted it through principal component analysis. We then calculated each type with an eigenvalue of at least 1.000. Finally, we examined the reasons for selecting the most agreeable or disagreeable items, centering on P samples with a high weight in each type.

## 3. Study Results

### 3.1. Result Analysis

As a result of the study, we derived five types of gratitude. The eigenvalues by type were: 10.0690 (Type 1); 5.2637 (Type 2); 2.7542 (Type 3); 1.6508 (Type 4), and; 1.3296 (Type 5). The cumulative variance was 0.4580 as shown in Table 4.

Table 5 shows the correlation coefficients showing similarity between individual types were as follows: 0.214 (Types 1 and 2); 0.367 (Types 1 and 3); 0.402 (Types 1 and 4); 0.300 (Types 1 and 5); 0.300 (Types 2 and 3); 0.441 (Types 2 and 4); 0.303 (Types 2 and 5); 0.300 (Types 2 and 5); 0.293 (Types 3 and 4); 0.208 (Types 3 and 5), and; 0.318 (Types 4 and 5).

We classified the factor weights and obtained the results shown in Table 6. In the case of Type 1, P10 showed the highest value at 1.8485; Type 2, P36 at 1.6604; Type 3, P27 at 1.6788; Type 4, P3 at 1.1445, and; Type 5, P38 at 0.9271. 

### 3.2. Perception Type Characteristics

#### 3.2.1. Type 1: Active Gratitude through Expression

Type 1 is a type that transcends the surrounding environment and conditions, and practices and expresses gratitude every moment. We named this type ‘active gratitude through expression’. This type showed the strongest agreement with Q36 (Being grateful makes me feel positive emotions, z = 1.91), followed by Q30 (I think it is important to express gratitude, z = 1.87), Q24 (I am grateful because I have a loving family, z = 1.47), and Q29 (I think I should be grateful every moment in everyday life, z = 1.36). Meanwhile, this type showed the strongest disagreement with Q38 (There are not many things for which I am grateful, z = −2.10), followed by Q23 (I feel grateful to God, z = −1.92), Q9 (Gratitude disappears when someone’s favor persists, z = −1.78), and Q7 (I feel repulsed by excessive expressions of gratitude, z = −1.44) as shown in Table 7.

P10 (1.8485), who had the highest factor weight among Type 1 samples, said, “A person who knows how to be grateful can accept the given reality and gain and enjoy as much as possible in the given environment. On the other hand, if a person does not know how to be grateful, they can hardly utilize the given environment properly. Since happiness is ultimately an emotion, being grateful can make the person’s thoughts stay in a positive place and eventually make the person happy”.

P11 (1.3467), who had the second highest factor weight among type 1 samples, said, “Although it is good to keep gratitude in my heart, I can feel my gratitude better when I express it firsthand. In addition, expressing gratitude makes me happier. Expressing gratitude can have positive effects on the other person so that the relationship can be improved and become closer”.

#### 3.2.2. Type 2: Passive Gratitude Depending on Conditions

Type 2 expresses gratitude passively depending on limited circumstances; we named this type ‘passive gratitude depending on conditions’. This type showed the strongest agreement with Q30 (I think it is important to express gratitude, z = 1.88)’, followed by Q18 (I am grateful when my friends help me, z = 1.80), Q24 (I am grateful because I have a loving family, z = 1.60), Q6 (I can be grateful for happy things, but cannot easily be grateful for painful things, z = 1.52), and Q11 (When life is hard, I hardly feel grateful, z = 1.37). Meanwhile, this type disagreed most with Q23 (I feel grateful to God, z = −2.16), followed by Q29 (I think I should be grateful every moment in everyday life, z = −1.65), Q14 (I am grateful when I successfully complete a task, z = −1.61), and Q26 (I am grateful to nature, such as air, water, and the sun, z = −1.58) as shown in Table 8.

P36 (1.6604), who had the highest factor weight among Type 2 samples, said, “If life is hard, I become pessimistic about my situation, and satisfaction and happiness levels drop significantly, so there are few things I should be grateful for, and it is difficult to feel gratitude. Being grateful is related to situations that have nothing to do with my will”.

P42 (1.6469), with the second highest factor weight among Type 2 samples, said, “It is not easy to feel gratitude because the environment around me is something I enjoy for granted, and since I am an atheist, I have never felt gratitude to God”.

#### 3.2.3. Type 3: Gratitude through Relationships

Type 3 feels gratitude through human relationships; we named it ‘gratitude through relationships’. This type showed the strongest agreement with Q24 (I am grateful because I have a loving family, z = 2.26), followed by Q13 (I am grateful because I met nice people at school, z = 1.87), Q17 (I feel grateful when I achieve good grades, z = 1.69), and Q18 (I am grateful when my friends help me, z = 1.67). However, this type strongly disagreed with items Q9 (Gratitude disappears when someone’s favor persists, z = −2.02), Q38 (There are not many things for which I am grateful, z = −1.85), Q2 (I think that if I keep expressing gratitude, I will easily create grateful situations, z = −1.20), and Q23 (I feel grateful to God, z = −1.15) as shown in Table 9.

P27 (1.1445), with the highest factor weight among the Type 3 samples, said, “I am grateful for the support of my parents and the good people I met while studying. I do not forget gratitude even if the favor persists, and I am grateful for many things”. P46 (1.1806), with the second highest factor weight, said, “I am grateful to my family members because they allow me to lean on them every time I am tired and exhausted and give me love. I am more grateful when someone’s favor persists”.

#### 3.2.4. Type 4: Active Gratitude through Internal Satisfaction

Type 4 feels gratitude through internal satisfaction; we named this type ‘active gratitude through internal satisfaction’. This type showed the strongest agreement with Q19 (I am grateful when I do something enjoyable, z = 2.09), followed by Q36 (Being grateful helps me to feel positive emotions, z = 1.66), Q18 (I am grateful, z = 1.49), and Q27 (I think gratitude is being satisfied with the current situation, z = 1.41). Meanwhile, the item with the strongest disagreement was Q15 (I am grateful when my professor respects me, z = −2.23), followed by Q38 (There are not many things for which I am grateful, z = −1.65), Q14 (I am grateful when I successfully complete a task, z = −1.48), and Q8 (I think a false expression of gratitude is not necessary, z = −1.42) as shown in Table 10.

P3 (1.6788), with the highest factor weight among the Type 4 samples, said, “I can feel gratitude when I do what I want to do, and gratitude makes me change positively”. P28 (0.7740), with the second highest factor weight, said, “I am grateful for the fact that there is something joyful and blissful to do, and I am more deeply grateful to a few people who positively affect me rather than being grateful to many people”.

#### 3.2.5. Type 5: Gratitude through Materials

Type 5 feels gratitude through visible materials; we named it ‘gratitude through materials’. This type showed the strongest agreement with Q3 (I think emotionally mature people express gratitude effectively, z = 1.85), followed by Q19 (I am grateful when I do something enjoyable, z = 1.73), Q7 (I feel repulsed by excessive expressions of gratitude, z = 1.66), and Q18 (I am grateful when my friends help me, z = 1.54). On the other hand, this type showed the strongest disagreement with Q23 (I feel grateful to God, z = −1.77), followed by Q8 (I think a false expression of gratitude is not necessary, z = −1.73), Q39 (I am grateful to myself because I think I am a grateful person, z = −1.58), Q38 (There are not many things for which I am grateful, z = −1.46), Q1 (Acts of gratitude easily spread to surrounding people, z = −1.46), and Q5 (I believe that the object of gratitude should be humans rather than materials or money, z = −1.35) as shown in Table 11.

P38 (0.9271), who had the highest factor weight among the Type 5 samples, said, “I don’t expect to receive favor without cost. The target of gratitude does not have to be a person. Since humans are multifaceted beings and change every moment, it is better to be grateful for materials than humans”.. P39 (0.6554), with the second highest factor weight, said, “I am not particularly grateful to air, water, sun, etc. because they cannot be owned by anyone and are free of charge”.

### 3.3. Consensus Items

We found two consensus items, i.e., items commonly agreed to by type, as shown in Table 12. Analyzing other items in consideration of consensus items helps understand each type’s characteristics. For instance, among the consensus items, no statement responded in a positive direction. On the other hand, consensus statements that responded in a negative direction were ‘Gratitude helps me to be more open to others’ and ‘I think a false expression of gratitude is not necessary’.

## 4. Discussion and Conclusions

Based on this study’s results, we divided college students’ perceptions of gratitude into five types: Type 1 active gratitude through expression; Type 2 passive gratitude depending on conditions; Type 3 gratitude through relationships; Type 4 gratitude through internal satisfaction, and; Type 5 gratitude through materials.

Eighteen students among the P samples belonged to Type 1. They believed that gratitude should be felt and expressed every moment in everyday life, and that there were many things for which they should be grateful. In addition, they thought that luck and good relationships come to those who know how to be grateful. For them, gratitude was a moment of blessing, and a state of mind of joy and thanks [35].

Thirteen students among the P samples belonged to Type 2. They were grateful for happy events, but found it difficult to be grateful in painful situations. Therefore, since they did not think they should be grateful for every moment, they considered gratitude as an emotion that occurs only under limited circumstances and conditions. Positive people do not easily fall into negative situations because they find positive aspects and interpret them beneficially, even when they encounter negative situations [36]. Conversely, Type 2 students could not find a positive aspect and connect it to gratitude in a negative situation. Type 2 students showed the lowest correlation (0.214) with Type 1 students, indicating that the types are relatively different.

Ten students among the P samples belonged to Type 3. They had good relationships with their loving families, nice people, and helpful friends; they also had the characteristic of perceiving gratitude when helped. In addition, they had a strong tendency to recognize the contributions of others, and respond with gratitude to positive results or experiences [8].

Three students among the P samples belonged to Type 4. They perceived gratitude through internal satisfaction and pleasure; they were grateful when they were satisfied with the current situation or when they felt joy. Since subjective well-being (SWB) and happiness positively affect their gratitude tendency [37], Type 4 students connected such internal satisfaction with gratitude. Type 4 students had the highest correlation (0.402) with Type 1 students.

Finally, we identified two students among the P samples belonging to Type 5. They believed they should be more grateful for money and materials than for people, but they do not feel gratitude for nature, such as air, water, and sun, because they believe nature is free. Gratitude by mutual exchange, one of the forms of gratitude described by Buck [38], is the gratitude the recipient and the giver feel thanks to the benefits of mutual exchange. Type 5 feels grateful for the benefits from mutual exchanges of material values, because this type attaches greater importance to material values.

We used GRAT (Gratitude, Resentment and Appreciation Test) to examine the characteristics of each type of perception of gratitude by gratitude factor (awareness of affluence, gratitude in daily life, and gratitude to others). We then calculated the sums of Z scores by type, classifying them as shown in Table 13.

We used the above classifications to examine the extent of each gratitude factor’s positive or negative influence by type through statistical values. We can sum Z-scores by applying weights to each research unit [39]. The sample size ($N_{i}$) for Z-score weight calculation in this study is 40, which is the number of Q samples. We obtained the total sample sizes ($N_{total}$) of the entire study by multiplying 40 × 9 (z-score × the number of statements for ‘awareness of affluence’), 40 × 13 (z-score × the number of statements for ‘gratitude in daily life’), and 40 × 18 (z-score × the number of statements for ‘gratitude to others’). Therefore, we can obtain the sum of the Z values by multiplying each Z value by $\frac{\sqrt{40}}{\sqrt{40*9}}$ ≒ 0.33, $\frac{\sqrt{40}}{\sqrt{40*13}}$ ≒ 0.28, and $\frac{\sqrt{40}}{\sqrt{40*18}}$ ≒ 0.24, which are weights by factor.

We referred to De Bakker et al.’s [38] paper for calculating the sum of the Z scores. We reverse-coded negative statements Q4, Q7, Q8, Q9, and Q11, so that the Z value became positive when all statements expressed gratitude positively.
$$Z_{meta}=\underset{i}{\sum }Z_{i\;}\;W_{i},\;\;W_{i}=\sqrt{N_{i}/N_{total}}\;$$

Concerning ‘awareness of affluence’ among the gratitude factors of GRAT, we found the following characteristics by type. We classified Type 1, ‘active gratitude through expression’, as one that best recognizes affluence because we obtained the highest sum of Z values in the case of ‘awareness of affluence’. We classified Type 2, ‘passive gratitude depending on conditions’, as a type that recognized affluence most poorly because we obtained the lowest sum of Z values in the case of ‘awareness of affluence’. In particular, the sum of Z values was negative only in the awareness of affluence. ‘Have focus’, a sub-factor of gratitude by Adler and Fagley [7], paid more attention to what one already had rather than what one lacked, and realizing the meanings of materials and things connected to or around the person [24]. Thus, Type 1 had more awareness and a sense of connection with the person’s things, while Type 2 did not have this awareness.

When we reviewed the characteristics by type relating to ‘gratitude in daily life’, we found Type 1 best perceives gratitude in daily life because we obtained the highest sum of the Z values in the case of ‘gratitude in daily life’. However, we obtained the lowest sum of Z values for Type 2, ‘passive gratitude depending on conditions’, indicating that this type perceived gratitude poorly in daily life. We also found Type 5, ‘gratitude through materials’, as a type that did not easily perceive gratitude in daily life, because the sum of Z values in the case of ‘gratitude in daily life’ was negative. People with a high gratitude disposition have developed the cognitive schema to see the positive aspects of various things they experience in daily life and be grateful for them [21]. Therefore, we could infer that Type 1, ‘active gratitude through expression’, had a well-developed cognitive schema to be grateful, while Type 2, ‘passive gratitude depending on conditions’, and Type 5, ‘gratitude through materials’, had a less developed cognitive schema to be grateful.

When we reviewed the characteristics by type regarding ‘gratitude to others’, we classified Type 3, ‘gratitude through relationships’, as the type that best perceives gratitude to others, because we obtained the highest sum of Z values in the case of ‘gratitude to others’. We classified Type 5, ‘gratitude through materials’, as a type that felt gratitude to others the most poorly. Type 4, ‘gratitude through internal satisfaction’, was also a type that does not easily feel gratitude towards others because the sum of Z values in the case of ‘gratitude to others’ was negative. The more stable attachment a person had, the higher the level of gratitude the person showed in forming relationships and interactions with others [40]. Therefore, Type 1, ‘gratitude through relationships’, formed stable attachments with others, while Type 5, ‘gratitude through materials’, and Type 4, ‘gratitude through internal satisfaction’, formed weak attachments.

Based on McCullough, Emmons, and Tsang’s [8] declaration that gratitude dispositions have intensity, frequency, span, and density, we can analyze the relationships between individual type and GRAT’s gratitude factors.

Type 1, ‘active gratitude through expression’, showed the highest sum of Z values in the case of ‘awareness of affluence’ and ‘gratitude in daily life’, and the second highest sum of Z values in the case of ‘gratitude to others’. This type had a gratitude disposition with high intensity and a wide range. Therefore, one should encourage this type to keep its attitude of being grateful as it is now.

Type 2, ‘passive gratitude depending on conditions’, was a type that generally did not feel gratitude strongly, and showed the lowest sum of Z values in the case of ‘awareness of affluence’ and ‘gratitude in daily life’. This type had a gratitude disposition with low intensity, frequency, and a narrow range. For this type of person, a program to promote the gratitude disposition [41], which increases gratitude and the contribution of others from their positive experiences, may be effective.

Type 3, ‘gratitude through relationships’, felt gratitude to others the most because this type showed the highest sum of Z values in the case of ‘gratitude to others,’ but generally low sum of Z scores in the case of ‘awareness of affluence’. Therefore, this type of person needs programs that promote a range of gratitude.

Type 4, ‘active gratitude through internal satisfaction’, showed the highest sum of Z scores in the case of ‘awareness of affluence’, but the sum of Z scores in the case of ‘gratitude to others’ was negative. For this type of person, a program that makes them perceive ‘gratitude to others’ may be effective.

Finally, Type 5, ‘gratitude through materials’, showed generally low sums of Z scores overall. In particular, the lowest sum of Z scores among the five types was in the case of ‘gratitude to others’. As with Type 2, a program that increases the overall gratitude disposition can be effective for this type.

As a result of this study, we identified differences in the degree to which one feels gratitude depending on situations among individual types. Individuals can feel gratitude only when they combine the positive aspect of the present moment and the experience of being grateful [7]. In addition, gratitude is a two-stage cognitive process. The first stage is perceiving positive results, while the second is perceiving that the cause of positive results is external [42]. Therefore, for a person to feel gratitude, they should feel positive emotions toward the object. Many studies show that people with a higher gratitude tendency have higher positive psychological capital [43,44,45,46]. Therefore, strong positivity for the current environment leads to high gratitude.

This study started by exploring an individual’s subjective inner world through the Q methodology, and classified individual preferences, emotions, ideals, and tastes; therefore, it can be said to be generating, rather than verifying, the hypothesis. This study quantitatively measured college students’ perception of gratitude and differentiated between types. The positive and negative perceptions of each type were quantitatively expressed, making the study useful for developing evaluation scales and items of experimental research in future gratitude research.

This study explored college students’ perceptions of gratitude through the Q methodology, but were are some limitations. First, whether each type derived with the Q methodology sufficiently reflected many individuals’ thoughts about gratitude is unclear. Through in-depth qualitative research on each individual’s experience of gratitude, it is necessary to analyze and study the internal experience and awareness of gratitude. To this end, it is essential to conduct in-depth research on gratitude felt by individuals through phenomenological research methods; these methods provide useful qualitative research methods related to gratitude. Secondly, there may be differences in the perception of gratitude between college students and other individuals. While carrying out this study, we felt that the objects and range of gratitude for college students were very narrow. Resultantly, it is questionable whether each type derived from the Q methodology can be classified similarly in other countries, because different individuals’ positive experiences and their results can affect perceptions of gratitude [8].

Therefore, we believe that future research is necessary to broaden the understanding and scope of gratitude; this research must be influenced by culture and customs. To do this, we need to diversify the range of study subjects by generation, gender, and country. This will facilitate a more comprehensive understanding of the thoughts and behaviors of that influence other individuals’ attitudes to gratitude.

## Figures and Tables

**Table 1 behavsci-13-00281-t001:** Study Process.

Stage	Content
Stage 1: Organization of Q population	Related literature surveys, interviews, questionnaire surveys
Stage 2: Selection of Q samples	Q statement sorting and supplementation, N = 40
Stage 3: Composition of P samples	College student group, N = 46
Stage 4: Reliability test	Securing reliability by test–retest
Stage 5: Q sorting and data analysis	Compulsory sorting by P samples and analyses by type

**Table 2 behavsci-13-00281-t002:** Q Statements.

Sorting of Gratitude	Q Statement
Gratitude to others	Q1. Acts of gratitude easily spread to surrounding people.
Gratitude in daily life	Q2. I think that if I keep expressing gratitude, I will easily create grateful situations.
Awareness of affluence	Q3. I think emotionally mature people express gratitude effectively.
Gratitude to others	Q4. I think many people do not know when they should be grateful.
Gratitude to others	Q5. I believe that the object of gratitude should be humans, rather than materials or money.
Gratitude in daily life	Q6. I can be grateful for happy things, but cannot easily be grateful for painful things.
Gratitude in daily life	Q7. I feel repulsed by excessive expressions of gratitude.
Gratitude in daily life	Q8. I think a false expression of gratitude is not necessary.
Gratitude to others	Q9. Gratitude disappears when someone’s favor persists.
Awareness of affluence	Q10. There are cases where I am grateful for my circumstances when I see people less fortunate than myself.
Gratitude in daily life	Q11. When life is hard, I hardly feel grateful.
Gratitude in daily life	Q12. I am grateful when I learn something I did not know through a class.
Gratitude to others	Q13. I am grateful because I met nice people at school.
Gratitude in daily life	Q14. I am grateful when I successfully complete a task.
Gratitude to others	Q15. I am grateful when my professor respects me.
Gratitude to others	Q16. I am grateful when I see someone sacrifices themself for others.
Gratitude in daily life	Q17. I feel grateful when I achieve good grades.
Gratitude to others	Q18. I am grateful when my friends help me.
Awareness of affluence	Q19. I am grateful when I do something enjoyable.
Gratitude to others	Q20. I am grateful when my parents praise me.
Gratitude in daily life	Q21. I am grateful because I am currently healthy.
Gratitude to others	Q22. I am grateful when I am helpful to someone.
Gratitude to others	Q23. I feel grateful to God.
Gratitude to others	Q24. I am grateful because I have a loving family.
Gratitude in daily life	Q25. I am grateful because I have food for every meal.
Gratitude in daily life	Q26. I am grateful to nature, such as air, water, and the sun.
Awareness of affluence	Q27. I think gratitude is being satisfied with the current situation.
Awareness of affluence	Q28. I think gratitude is something I can give for free.
Gratitude in daily life	Q29. I think I should be grateful for every moment in everyday life.
Gratitude to others	Q30. I think it is important to express gratitude.
Awareness of affluence	Q31. I think luck comes to those who know how to be grateful.
Gratitude in daily life	Q32. Gratitude gives me the strength to overcome difficult situations.
Gratitude to others	Q33. I think that if I express gratitude effectively, I will be helped by surrounding people.
Gratitude to others	Q34. I think expressing gratitude creates good relationships.
Gratitude to others	Q35. When I feel gratitude, I find the value of the other person.
Awareness of affluence	Q36. Being grateful helps me to feel positive emotions.
Gratitude to others	Q37. Gratitude helps me to be more open to others.
Awareness of affluence	Q38. There are not many things for which I am grateful.
Gratitude in daily life	Q39. I am grateful to myself because I think I am a grateful person.
Gratitude to others	Q40. I am grateful to so many people.

**Table 3 behavsci-13-00281-t003:** Correlation between first and second Q sorting.

Participant	Time of Re-Sorting	Correlation Coefficient (r)	Average Correlation Coefficient (r)
P15	7 days later	0.764 **	0.682 **
P31	7 days later	0.551 **	
P45	7 days later	0.733 **	

** *p* < 0.01.

**Table 4 behavsci-13-00281-t004:** Eigenvalues and explanatory variances in the classification of five types.

Content	I	II	III	IV	V
Chosen eigenvalues	10.0690	5.2637	2.7542	1.6508	1.3296
Total variance	0.2189	0.1144	0.0599	0.0359	0.0289
Cumulative	0.2189	0.3333	0.3932	0.4291	0.4580

**Table 5 behavsci-13-00281-t005:** Correlation coefficients between types.

Type	I	II	III	IV	V
I	1.000				
II	0.214	1.000			
III	0.367	0.441	1.000		
IV	0.402	0.303	0.293	1.000	
V	0.300	0.300	0.208	0.318	1.000

**Table 6 behavsci-13-00281-t006:** Subjects and Factor Weights by Type.

No.	Factor Loading	Gender	Grade
**Type 1, N = 18**			
P2	0.7697	Male	1
P5	1.2560	Female	4
P6	0.3081	Female	3
P7	1.1970	Female	3
P10	1.8485	Male	4
P11	1.3467	Male	1
P13	0.7843	Male	1
P17	0.6529	Female	3
P18	0.6695	Female	4
P23	0.8972	Female	2
P25	0.4360	Male	4
P26	0.2003	Female	3
P29	0.9928	Female	4
P30	1.0245	Male	1
P32	0.9976	Male	2
P37	0.9500	Female	4
P44	0.7452	Female	4
P45	0.4576	Female	1
**Type 2, N = 13**			
P1	1.6005	Male	4
P8	1.4853	Female	3
P12	1.1956	Male	1
P15	0.3727	Male	1
P19	0.3937	Male	1
P20	0.7384	Female	2
P21	0.5401	Male	4
P24	0.8575	Female	3
P31	0.7293	Female	2
P36	1.6604	Male	3
P41	0.7684	Female	2
P42	1.6469	Male	3
P43	1.1157	Female	4
**Type 3, N = 10**			
P4	0.7491	Male	1
P9	0.9088	Male	4
P14	0.8021	Female	1
P16	0.4421	Female	1
P22	0.8530	Female	2
P27	1.6788	Female	2
P34	0.5999	Female	1
P35	0.7024	Female	2
P40	0.6568	Female	3
P46	1.1806	Male	1
**Type 4, N = 3**			
P3	1.1445	Male	4
P28	0.7740	Female	3
P33	0.6928	Male	2
**Type 5, N = 2**			
P38	0.9271	Female	4
P39	0.6554	Male	4

**Table 7 behavsci-13-00281-t007:** Statements and standard scores (at least ±1.00) of Type 1.

No.	Statement	Standard Score
36	Being grateful helps me to feel positive emotions.	1.91
30	I think it is important to express gratitude.	1.87
24	I am grateful because I have a loving family.	1.47
29	I think I should be grateful for every moment in everyday life.	1.36
31	I think luck comes to those who know how to be grateful.	1.34
34	I think expressing gratitude creates good relationships.	1.26
11	When life is hard, I hardly feel grateful.	−1.26
8	I think a false expression of gratitude is not necessary.	−1.35
7	I feel repulsed by excessive expressions of gratitude.	−1.44
9	Gratitude disappears when someone’s favor persists.	−1.78
23	I feel grateful to God.	−1.92
38	There are not many things for which I am grateful.	−2.10

**Table 8 behavsci-13-00281-t008:** Statements and standard scores (at least ±1.00) of Type 2.

No.	Statement	Standard Score
30	I think it is important to express gratitude.	1.88
18	I am grateful when my friends help me.	1.80
24	I am grateful because I have a loving family.	1.60
6	I can be grateful for happy things, but cannot easily be grateful for painful things.	1.52
11	When life is hard, I hardly feel grateful.	1.37
34	I think expressing gratitude creates good relationships.	1.28
21	I am grateful because I am currently healthy.	1.07
32	Gratitude gives me the strength to overcome difficult situations.	−1.09
31	I think luck comes to those who know how to be grateful.	−1.14
2	I think that if I keep expressing gratitude, I will easily create grateful situations.	−1.14
39	I am grateful to myself because I think I am a grateful person.	−1.16
26	I am grateful to nature, such as air, water, and the sun.	−1.58
14	I am grateful when I successfully complete a task.	−1.61
29	I think I should be grateful for every moment in everyday life.	−1.65
23	I feel grateful to God.	−2.16

**Table 9 behavsci-13-00281-t009:** Statements and standard scores (at least ±1.00) of Type 3.

No.	Statement	Standard Score
24	I am grateful because I have a loving family.	2.26
13	I am grateful because I met nice people at school.	1.87
17	I feel grateful when I achieve good grades.	1.69
18	I am grateful when my friends help me.	1.67
21	I am grateful because I am currently healthy.	1.47
19	I am grateful when I do something enjoyable.	1.44
22	I am grateful when I am helpful to someone.	1.25
20	I am grateful when my parents praise me.	1.16
23	I feel grateful to God.	−1.15
2	I think that if I keep expressing gratitude, I will easily create grateful situations.	−1.20
38	There are not many things for which I am grateful.	−1.85
9	Gratitude disappears when someone’s favor persists.	−2.02

**Table 10 behavsci-13-00281-t010:** Statements and Standard Scores (at least ±1.00) of Type 4.

No.	Statement	Standard Score
19	I am grateful when I do something enjoyable.	2.09
36	Being grateful helps me to feel positive emotions.	1.66
18	I am grateful when my friends help me.	1.49
27	I think gratitude is being satisfied with the current situation.	1.41
2	I think that if I keep expressing gratitude, I will easily create grateful situations.	1.34
29	I think I should be grateful for every moment of everyday life.	1.08
16	I am grateful when I see someone sacrifices themself for others.	1.03
12	I am grateful when I learn something I did not know through a class.	−1.04
32	Gratitude gives me the strength to overcome difficult situations.	−1.39
40	I am grateful to so many people.	−1.41
8	I think a false expression of gratitude is not necessary.	−1.42
14	I am grateful when I successfully complete a task.	−1.48
38	There are not many things for which I am grateful.	−1.65
15	I am grateful when my professor respects me.	−2.23

**Table 11 behavsci-13-00281-t011:** Statements and standard scores (at least ±1.00) of type 5.

No.	Statement	Standard Score
3	I think emotionally mature people express gratitude effectively.	1.85
19	I am grateful when I do something enjoyable.	1.73
7	I feel repulsed by excessive expressions of gratitude.	1.66
18	I am grateful when my friends help me.	1.54
36	Being grateful helps me to feel positive emotions.	1.39
16	I am grateful when I see someone sacrifices themself for others.	1.23
11	When life is hard, I hardly feel grateful.	1.19
20	I am grateful when my parents praise me.	−1.00
31	I think luck comes to those who know how to be grateful.	−1.11
5	I believe that the object of gratitude should be humans, rather than materials or money.	−1.35
1	Acts of gratitude easily spread to surrounding people.	−1.46
38	There are not many things for which I am grateful.	−1.46
39	I am grateful to myself because I think I am a grateful person.	−1.58
8	I think a false expression of gratitude is not necessary.	−1.73
23	I feel grateful to God.	−1.77

**Table 12 behavsci-13-00281-t012:** Consensus items of each type.

No.	Statement	Standard Score
37	Gratitude helps me to be more open to others.	−0.34
8	I think a false expression of gratitude is not necessary.	−1.24

**Table 13 behavsci-13-00281-t013:** Sums of Z-Scores (Ranking) of Q Statement Classified by GRAT’s Gratitude Factors.

	Classification by GRAT	Awareness of Affluence	Gratitude in Daily Life	Gratitude to Others
Type				
Type 1: Active gratitude through expression		2.24 (1)	1.18 (1)	1.3 (2)
Type 2: Passive gratitude depending on conditions		−0.07 (5)	−1.66 (5)	0.65 (3)
Type 3: Gratitude through relationships		0.73 (4)	0.50 (3)	2.09 (1)
Type 4: Gratitude through internal satisfaction		2.08 (2)	0.64 (2)	−0.17 (4)
Type 5: Gratitude through materials		1.68 (3)	−0.84 (4)	−0.53 (5)

## Data Availability

Data from the study are available upon request.

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
