# Peer review of "A Study on South Korean College Students’ Perceptions of Gratitude"

_behavsci, 2023, doi:10.3390/bs13040281_

Round 1

Reviewer 1 Report

Thank you for allowing me to review this interesting study. Overall, the study has raised a very interesting point of discussion. I think this study has provided novel findings in this area, allowing readers to think more deeply about what is going on.

First of all, I would like to share the need to carry out works like the one you present. They are necessary for the advancement of science in the field they study. The objective of the manuscript is clear and consistent. The study has been an interesting reading, it is necessary to know the reality of the sector on which the work emphasizes.

The abstract includes the necessary elements: background with purpose (objective) of the study, methods, results, main conclusions without exaggeration.

In the introduction, sufficient ordered references of the publications considered key are indicated, with significant and sufficient evidence. I find the literature review very interesting.

Likewise, reasons are highlighted that justify the importance in a broad context and the current state of the subject investigated. The study is clearly defined and indicates the intention and meaning of the work. The target to be tested in the study is recorded. The text is understandable and makes clear the main objective of the work and the main conclusions.

In relation to the material and methods, say that the study is described in detail. In addition to the methods, the intervention requirements are indicated in sufficient detail.

Overall, it is a very interesting manuscript, despite some suggested questions to improve the manuscript and study findings.

It is recommended to include the limitations, more specifically, the difficulties that the authors have detected. If you consider that your findings can be extrapolated to other countries, among other issues, and I would like to know if you believe that the sport discipline to which you belong and the level at which you practice it may present significant differences depending on the results obtained. Also, maybe I would suggest incorporating sports satisfaction and/or facing competition as a keyword, because many of them compete.

Author Response

Dear Reviewer,

Thank you for taking the time to provide us with your feedback. We appreciate your input and have carefully reviewed your comments. However, we have noticed that some of the points you raised were not directly related to the subject matter of our paper, such as sport discipline and sports satisfaction. As a result, we have revised the paper based on the feedback from the second reviewer. Thank you for your understanding.

Best regards,

Yucheon Kim

Reviewer 2 Report

The methodology of Q Methodology was conducted well and is at state of the art.

I have several aspects of the current manuscript I like to highlight as critical, and which needed to be improved:

(1) the methodology is introduced but I am missing a precise differentiation to hypotheses testing methodology. According to critical rationalism, we have to differentiate clearly between deductive Methodology which is theory driven and hypotheses and testing and inductive methodology, which is  hypotheses generating. So together with all qualitative methods the Q Methodology is clearly inductive. This should be explained more explicitly to the readers in the introduction.

Further all results by the Q- Methodology are not tested in the understanding of critical rationalism. They are just generated classification based on subjective assumption without any falsification/verification record.

In this sense the classification are yet untested hypotheses about the structure of the investigated construct.

The generation of new hypotheses or new classifications of a construct are important but o my understanding this is necessary to highlight that they are not yet tested to avoid misinterpretation.

The aspect of (1) is associated with my further aspects (2) to (4), which should however addressed in the discussion part.

(2) What aspects could be overseen by the students? What could be contributed by other individuals? both aspects are missing the discussion section. Please add a short passage discussing this important limitation. The statement " Therefore, it is necessary to diversify the range of study subjects for gratitude perception to understand the differences by generation, gender, and country" is correct but is too general. Please specify.

Forthermore the following aspects should also be discussed in the discussion/conclusion section:

(3) what type of future research should be conducted next? theory driven work or empirical quantitative testing.

The items found by Q methodology can be the starting point for creating a quantitative scale to measure different types of gratitude and / or levels of gratitude.  Please explain how your work can lead to the next steps as a scale. The authors already stated: ""Therefore, we think that future research is necessary to broaden the understanding and scope of college students’ gratitude and conduct further studies to provide quantitative evidence."" Agai,n this is too general and should me more specified.

 (4) could the classification be relevant outside of South Korea. Please explain why yes or no.

Author Response

Dear Reviewer,

Thank you for your valuable feedback and allowing us to submit the revised draft of the manuscript “A Study on South Korean College Students’ Perceptions of Gratitude” for behavioral sciences. We have put a lot of effort into revising manuscript according to the comments made by the reviewers.

Best regards,

Yucheon Kim
